# VICReg: Variance-Invariance-Covariance Regularization for Self-Supervised Learning

**Adrien Bardes**[1,2]                **Jean Ponce**[2,4]                **Yann LeCun**[1,3,4]

[1]Facebook AI Research
[2]Inria, École normale supérieure, CNRS, PSL Research University
[3]Courant Institute, New York University
[4]Center for Data Science, New York University

## Abstract

Recent self-supervised methods for image representation learning maximize the agreement between embedding vectors produced by encoders fed with different views of the same image. The main challenge is to prevent a *collapse* in which the encoders produce constant or non-informative vectors. We introduce VICReg (Variance-Invariance-Covariance Regularization), a method that explicitly avoids the collapse problem with two regularizations terms applied to both embeddings separately: (1) a term that maintains the variance of each embedding dimension above a threshold, (2) a term that decorrelates each pair of variables. Unlike most other approaches to the same problem, VICReg does *not* require techniques such as: weight sharing between the branches, batch normalization, feature-wise normalization, output quantization, stop gradient, memory banks, etc., and achieves results on par with the state of the art on several downstream tasks. In addition, we show that our variance regularization term stabilizes the training of other methods and leads to performance improvements.

## 1 Introduction

Self-supervised representation learning has made significant progress over the last years, almost reaching the performance of supervised baselines on many downstream tasks Bachman et al. (2019); Misra & Maaten (2020); He et al. (2020); Tian et al. (2020); Caron et al. (2020); Grill et al. (2020); Chen & He (2020); Gidaris et al. (2021); Zbontar et al. (2021). Several recent approaches rely on a *joint embedding architecture* in which two networks are trained to produce similar embeddings for different views of the same image. A popular instance is the Siamese network architecture Bromley et al. (1994), where the two networks share the same weights. The main challenge with joint embedding architectures is to prevent a *collapse* in which the two branches ignore the inputs and produce identical and constant output vectors. There are two main approaches to preventing collapse: contrastive methods and information maximization methods. Contrastive Bromley et al. (1994); Chopra et al. (2005); He et al. (2020); Hjelm et al. (2019); Chen et al. (2020a) methods tend to be costly, require large batch sizes or memory banks, and use a loss that explicitly pushes the embeddings of dissimilar images away from each other. They often require a mining procedure to search for offending dissimilar samples from a memory bank He et al. (2020) or from the current batch Chen et al. (2020a). Quantization-based approaches Caron et al. (2020; 2018) force the embeddings of different samples to belong to different clusters on the unit sphere. Collapse is prevented by ensuring that the assignment of samples to clusters is as uniform as possible. A similarity term encourages the cluster assignment score vectors from the two branches to be similar. More recently, a few methods have appeared that do not rely on contrastive samples or vector quantization, yet produce high-quality representations, for example BYOL Grill et al. (2020) and SimSiam Chen & He (2020). They exploit several tricks: batch-wise or feature-wise normalization, a "momentum encoder" in which the parameter vector of one branch is a low-pass-filtered version of the parameter vector of the other branch Grill et al. (2020); Richemond et al. (2020), or a stop-gradient operation in one of the branches Chen & He (2020). The dynamics of learning in these methods, and how they avoid collapse, is not fully understood, although theoretical and empirical studies point to the crucial importance of batch-wise or feature-wise normalization Richemond et al. (2020); Tian et al. (2021). Finally, an

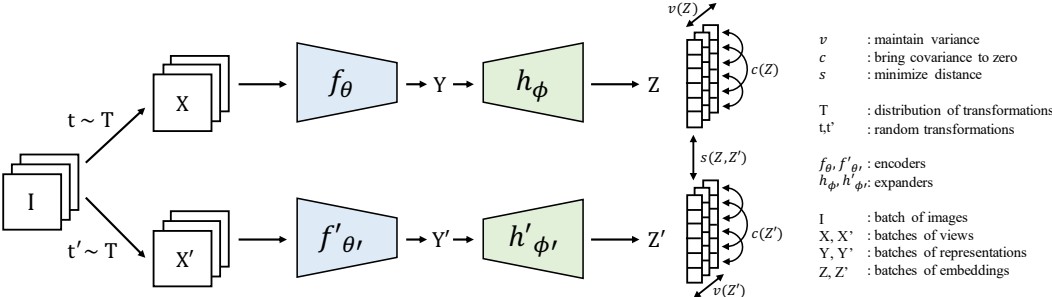

Figure 1: **VICReg: joint embedding architecture with variance, invariance and covariance regularization.** Given a batch of images $I$, two batches of different views $X$ and $X'$ are produced and are then encoded into representations $Y$ and $Y'$. The representations are fed to an expander producing the embeddings $Z$ and $Z'$. The distance between two embeddings from the same image is minimized, the variance of each embedding variable over a batch is maintained above a threshold, and the covariance between pairs of embedding variables over a batch are attracted to zero, decorrelating the variables from each other. Although the two branches do not require identical architectures nor share weights, in most of our experiments, they are Siamese with shared weights: the encoders are ResNet-50 backbones with output dimension 2048. The expanders have 3 fully-connected layers of size 8192.

alternative class of collapse prevention methods relies on maximizing the information content of the embedding Zbontar et al. (2021); Ermolov et al. (2021). These methods prevent *informational collapse* by decorrelating every pair of variables of the embedding vectors. This indirectly maximizes the information content of the embedding vectors. The Barlow Twins method drives the normalized cross-correlation matrix of the two embeddings towards the identity Zbontar et al. (2021), while the Whitening-MSE method whitens and spreads out the embedding vectors on the unit sphere Ermolov et al. (2021).

## 2 VICReg: INTUITION

We introduce VICReg (Variance-Invariance-Covariance Regularization), a self-supervised method for training joint embedding architectures based on the principle of preserving the information content of the embeddings. The basic idea is to use a loss function with three terms:

- **Invariance**: the mean square distance between the embedding vectors.

- **Variance**: a hinge loss to maintain the standard deviation (over a batch) of each variable of the embedding above a given threshold. This term forces the embedding vectors of samples within a batch to be different.

- **Covariance**: a term that attracts the covariances (over a batch) between every pair of (centered) embedding variables towards zero. This term decorrelates the variables of each embedding and prevents an *informational collapse* in which the variables would vary together or be highly correlated.

Variance and Covariance terms are applied to both branches of the architecture separately, thereby preserving the information content of each embedding at a certain level and preventing informational collapse independently for the two branches. The main contribution of this paper is the Variance preservation term, which explicitly prevents a collapse due to a shrinkage of the embedding vectors towards zero. The Covariance criterion is borrowed from the Barlow Twins method and prevents informational collapse due to redundancy between the embedding variables Zbontar et al. (2021). VICReg is more generally applicable than most of the aforementioned methods because of fewer constraints on the architecture. In particular, VICReg:

- does not require that the weights of the two branches be shared, not that the architectures be identical, nor that the inputs be of the same nature;

- does not require a memory bank, nor contrastive samples, nor a large batch size;
- does not require batch-wise nor feature-wise normalization; and
- does not require vector quantization nor a predictor module.

Other methods require asymmetric stop gradient operations, as in SimSiam Chen & He (2020), weight sharing between the two branches as in classical Siamese nets, or weight sharing through exponential moving average dampening with stop gradient in one branch, as in BYOL and MoCo He et al. (2020); Grill et al. (2020); Chen et al. (2020c), large batches of contrastive samples, as in SimCLR Chen et al. (2020a), or batch-wise and/or feature-wise normalization Caron et al. (2020); Grill et al. (2020); Chen & He (2020); Zbontar et al. (2021); Ermolov et al. (2021). One of the most interesting feature of VICReg is the fact that the two branches are not required to share the same parameters, architecture, or input modality. This opens the door to the use of non-contrastive self-supervised joint-embedding for multi-modal signals, such as video and audio. We demonstrate the effectiveness of the proposed approach by evaluating the representations learned with VICReg on several downstream image recognition tasks including linear head and semi-supervised evaluation protocols for image classification on ImageNet Deng et al. (2009), and other classification, detection, instance segmentation, and retrieval tasks. Furthermore, we show that incorporating variance preservation into other self-supervised joint-embedding methods yields better training stability and performance improvement on downstream tasks. More generally, we show that VICReg is an explicit and effective, yet simple method for preventing collapse in self-supervised joint-embedding learning.

## 3 RELATED WORK

**Contrastive learning.** In contrastive SSL methods applied to joint embedding architectures, the output embeddings for a sample and its distorted version are brought close to each other, while other samples and their distortions are pushed away. The method is most often applied to Siamese architectures in which the two branches have identical architectures and share weights Misra & Maaten (2020); He et al. (2020); Bromley et al. (1994); Hjelm et al. (2019); Chen et al. (2020a;c); Hadsell et al. (2006); Ye et al. (2019); Wu et al. (2018); van den Oord et al. (2018); Chen et al. (2020b). Many authors use the InfoNCE loss van den Oord et al. (2018) in which the repulsive force is larger for contrastive samples that are closer to the reference. While these methods yield good performance, they require large amounts of contrastive pairs in order to work well. These contrastive pairs can be sampled from a memory bank as in MoCo He et al. (2020), or given by the current batch of data as in SimCLR Chen et al. (2020a), with a significant memory footprint. This downside of contrastive methods motivates a search for alternatives.

**Clustering methods.** Instead of viewing each sample as its own class, clustering-based methods group them into clusters based on some similarity measure Caron et al. (2020; 2018); Bautista et al. (2016); Yang et al. (2016); Xie et al. (2016); Huang et al. (2019); Zhuang et al. (2019); Caron et al. (2019); Asano et al. (2020); Yan et al. (2020). DeepCluster Caron et al. (2018) uses $k$-means assignments of representations from previous iterations as pseudo-labels for the new representations, which requires an expensive clustering phase done asynchronously, and makes the method hard to scale up. SwAV Caron et al. (2020) mitigates this issue by learning the clusters online while maintaining a balanced partition of the assignments through the Sinkhorn-Knopp transform Cuturi (2013). These clustering approaches can be viewed as contrastive learning at the level of clusters which still requires a lot of negative comparisons to work well.

**Distillation methods.** Recent proposals such as BYOL, SimSiam, OBoW and variants Grill et al. (2020); Chen & He (2020); Gidaris et al. (2021); Richemond et al. (2020); Gidaris et al. (2020) have shown that collapse can be avoided by using architectural tricks inspired by knowledge distillation Hinton et al. (2015). These methods train a student network to predict the representations of a teacher network, for which the weights are a running average of the student network's weights Grill et al. (2020), or are shared with the student network, but no gradient is back-propagated through the teacher Chen & He (2020). These methods are effective, but there is no clear understanding of why and how they avoid collapse. Alternatively, the images can be represented as bags of word over a dictionary of visual features, which effectively prevents collapse. In OBoW Gidaris et al. (2020) and Gidaris et al. (2021) the dictionary is obtained by off-line or on-line clustering. By contrast, our method explicitly prevents collapse in the two branches independently, which removes

the requirement for shared weights and identical architecture, opening the door to the application of joint-embedding SSL to multi-modal signals.

**Information maximization methods.** A principle to prevent collapse is to maximize the information content of the embeddings. Two such methods were recently proposed: W-MSE Ermolov et al. (2021) and Barlow Twins Zbontar et al. (2021). In W-MSE, an extra module transforms the embeddings into the eigenspace of their covariance matrix (whitening or Karhunen-Loève transform), and forces the vectors thereby obtained to be uniformly distributed on the unit sphere. In Barlow Twins, a loss term attempts to make the normalized cross-correlation matrix of the embedding vectors from the two branches to be close to the identity. Both methods attempt to produce embedding variables that are decorrelated from each other, thus preventing an *informational collapse* in which the variables carry redundant information. Because all variables are normalized over a batch, there is no incentive for them to shrink nor expand. This seems to sufficient to prevent collapse. Our method borrows the decorrelation mechanism of Barlow Twins. But it includes an explicit variance-preservation term for each variable of the two embeddings and thus does not require any normalization.

## 4 VICREG: DETAILED DESCRIPTION

VICReg follows recent trends in self-supervised learning Caron et al. (2020); Grill et al. (2020); Chen & He (2020); Zbontar et al. (2021); Chen et al. (2020a) and is based on a *joint embedding architecture*. Contrary to many previous approaches, our architecture may be completely symmetric or completely asymmetric with no shared structure or parameters between the two branches. In most of our experiments, we use a Siamese net architecture in which the two branches are identical and share weights. Each branch consists of an *encoder* $f_\theta$ that outputs the representations (used for downstream tasks), followed by an *expander* $h_\phi$ that maps the representations into an embedding space where the loss function will be computed. The role of the expander is twofold: (1) eliminate the information by which the two representations differ, (2) expand the dimension in a non-linear fashion so that decorrelating the embedding variables will reduce the dependencies (not just the correlations) between the variables of the representation vector. The loss function uses a term $s$ that learns invariance to data transformations and is regularized with a variance term $v$ that prevents norm collapse and a covariance term $c$ that prevents informational collapse by decorrelating the different dimensions of the vectors. After pretraining, the expander is discarded and the representations of the encoder are used for downstream tasks.

### 4.1 METHOD

Given an image $i$ sampled from a dataset $\mathcal{D}$, two transformations $t$ and $t'$ are sampled from a distribution $\mathcal{T}$ to produce two different views $x = t(i)$ and $x' = t'(i)$ of $i$. These transformations are random crops of the image, followed by color distortions. The distribution $\mathcal{T}$ is described in Appendix C. The views $x$ and $x'$ are first encoded by $f_\theta$ into their *representations* $y = f_\theta(x)$ and $y' = f_\theta(x')$, which are then mapped by the expander $h_\phi$ onto the *embeddings* $z = h_\phi(y)$ and $z' = h_\phi(y')$. The loss is computed at the embedding level on $z$ and $z'$.

We describe here the variance, invariance and covariance terms that compose our loss function. The images are processed in batches, and we denote $Z = [z_1, \ldots, z_n]$ and $Z' = [z'_1, \ldots, z'_n]$ the two batches composed of $n$ vectors of dimension $d$, of embeddings coming out of the two branches of the siamese architecture. We denote by $z^j$ the vector composed of each value at dimension $j$ in all vectors in $Z$. We define the variance regularization term $v$ as a hinge function on the standard deviation of the embeddings along the batch dimension:

$$v(Z) = \frac{1}{d} \sum_{j=1}^{d} \max(0, \gamma - S(z^j, \epsilon)), \tag{1}$$

where $S$ is the regularized standard deviation defined by:

$$S(x, \epsilon) = \sqrt{\text{Var}(x) + \epsilon}, \tag{2}$$

$\gamma$ is a constant target value for the standard deviation, fixed to $1$ in our experiments, $\epsilon$ is a small scalar preventing numerical instabilities. This criterion encourages the variance inside the current

batch to be equal to $\gamma$ along each dimension, preventing collapse with all the inputs mapped on the same vector. Using the standard deviation and not directly the variance is crucial. Indeed, if we take $S(x) = \text{Var}(x)$ in the hinge function, the gradient of $S$ with respect to $x$ becomes close to 0 when $x$ is close to $\bar{x}$. In this case, the gradient of $v$ also becomes close to 0 and the embeddings collapse. We define the covariance matrix of $Z$ as:

$$C(Z) = \frac{1}{n-1} \sum_{i=1}^{n} (z_i - \bar{z})(z_i - \bar{z})^T, \quad \text{where} \quad \bar{z} = \frac{1}{n} \sum_{i=1}^{n} z_i. \tag{3}$$

Inspired by Barlow Twins Zbontar et al. (2021), we can then define the covariance regularization term $c$ as the sum of the squared off-diagonal coefficients of $C(Z)$, with a factor $1/d$ that scales the criterion as a function of the dimension:

$$c(Z) = \frac{1}{d} \sum_{i \neq j} [C(Z)]_{i,j}^2. \tag{4}$$

This term encourages the off-diagonal coefficients of $C(Z)$ to be close to 0, decorrelating the different dimensions of the embeddings and preventing them from encoding similar information. Decorrelation at the embedding level ultimately has a decorrelation effect at the representation level, which is a non trivial phenomenon that we study in Appendix D. We finally define the invariance criterion $s$ between $Z$ and $Z'$ as the mean-squared euclidean distance between each pair of vectors, without any normalization:

$$s(Z, Z') = \frac{1}{n} \sum_i \|z_i - z_i'\|_2^2. \tag{5}$$

The overall loss function is a weighted average of the invariance, variance and covariance terms:

$$\ell(Z, Z') = \lambda s(Z, Z') + \mu[v(Z) + v(Z')] + \nu[c(Z) + c(Z')], \tag{6}$$

where $\lambda$, $\mu$ and $\nu$ are hyper-parameters controlling the importance of each term in the loss. In our experiments, we set $\nu = 1$ and perform a grid search on the values of $\lambda$ and $\mu$ with the base condition $\lambda = \mu > 1$. The overall objective function taken on all images over an unlabelled dataset $\mathcal{D}$ is given by:

$$\mathcal{L} = \sum_{I \in \mathcal{D}} \sum_{t,t' \sim \mathcal{T}} \ell(Z^I, Z'^I), \tag{7}$$

where $Z^I$ and $Z'^I$ are the batches of embeddings corresponding to the batch of images $I$ transformed by $t$ and $t'$. The objective is minimized for several epochs, over the encoder parameters $\theta$ and expander parameters $\phi$. We illustrate the architecture and loss function of VICReg in Figure 1.

## 4.2 IMPLEMENTATION DETAILS

Implementation details for pretraining with VICReg on the 1000-classes ImagetNet dataset without labels are as follows. Coefficients $\lambda$ and $\mu$ are 25 and $\nu$ is 1 in Eq. (6), and $\epsilon$ is 0.0001 in Eq. (1). We give more details on how we choose the coefficients of the loss function in Appendix D.4. The encoder network $f_\theta$ is a standard ResNet-50 backbone He et al. (2016) with 2048 output units. The expander $h_\phi$ is composed of two fully-connected layers with batch normalization (BN) Ioffe & Szegedy (2015) and ReLU, and a third linear layer. The sizes of all 3 layers were set to 8192. As with Barlow Twins, performance improves when the size of the expander layers is larger than the dimension of the representation. The impact of the expander dimension on performance is studied in Appendix D. The training protocol follows those of BYOL and Barlow Twins: LARS optimizer You et al. (2017); Goyal et al. (2017) run for 1000 epochs with a weight decay of $10^{-6}$ and a learning rate $lr = batch\_size/256 \times base\_lr$, where $batch\_size$ is set to 2048 by default and $base\_lr$ is a base learning rate set to 0.2. The learning rate follows a cosine decay schedule Loshchilov & Hutter (2017), starting from 0 with 10 warmup epochs and with final value of 0.002.

## 5 RESULTS

In this section, we evaluate the representations obtained after self-supervised pretraining of a ResNet-50 He et al. (2016) backbone with VICReg during 1000 epochs, on the training set of ImageNet, using the training protocol described in section 4. We also pretrain on pairs of image and text data and evaluate on retrieval tasks on the MS-COCO dataset.

Table 1: **Evaluation on ImageNet.** Evaluation of the representations obtained with a ResNet-50 backbone pretrained with VICReg on: (1) linear classification on top of the frozen representations from ImageNet; (2) semi-supervised classification on top of the fine-tuned representations from 1% and 10% of ImageNet samples. We report Top-1 and Top-5 accuracies (in %). Top-3 best self-supervised methods are underlined.

| Method | Linear | | Semi-supervised | | | |
| | Top-1 | Top-5 | Top-1 | | Top-5 | |
| | | | 1% | 10% | 1% | 10% |
| --- | --- | --- | --- | --- | --- | --- |
| Supervised | 76.5 | - | 25.4 | 56.4 | 48.4 | 80.4 |
| MoCo He et al. (2020) | 60.6 | - | - | - | - | - |
| PIRL Misra & Maaten (2020) | 63.6 | - | - | - | 57.2 | 83.8 |
| CPC v2 Hénaff et al. (2019) | 63.8 | - | - | - | - | - |
| CMC Tian et al. (2019) | 66.2 | - | - | - | - | - |
| SimCLR Chen et al. (2020a) | 69.3 | 89.0 | 48.3 | 65.6 | 75.5 | 87.8 |
| MoCo v2 Chen et al. (2020c) | 71.1 | - | - | - | - | - |
| SimSiam Chen & He (2020) | 71.3 | - | - | - | - | - |
| SwAV Caron et al. (2020) | 71.8 | - | - | - | - | - |
| InfoMin Aug Tian et al. (2020) | 73.0 | 91.1 | - | - | - | - |
| OBoW Gidaris et al. (2021) | 73.8 | - | - | - | 82.9 | 90.7 |
| BYOL Grill et al. (2020) | 74.3 | 91.6 | 53.2 | 68.8 | 78.4 | 89.0 |
| SwAV (w/ multi-crop) Caron et al. (2020) | 75.3 | - | 53.9 | 70.2 | 78.5 | 89.9 |
| Barlow Twins Zbontar et al. (2021) | 73.2 | 91.0 | 55.0 | 69.7 | 79.2 | 89.3 |
| VICReg (ours) | 73.2 | 91.1 | 54.8 | 69.5 | 79.4 | 89.5 |

## 5.1 EVALUATION ON IMAGENET

Following the ImageNet Deng et al. (2009) linear evaluation protocol, we train a linear classifier on top of the frozen representations of the ResNet-50 backbone pretrained with VICReg. We also evaluate the performance of the backbone when fine-tuned with a linear classifier on a subset of ImageNet's training set using 1% or 10% of the labels, using the split of Chen et al. (2020a). We give implementation details about the optimization procedure for these tasks in Appendix C. We have applied the training procedure described in section 4 with three different random initialization. The numbers reported in Table 1 for VICReg are the mean scores, and we have observed that the difference between worse and best run is lower than 0.1% accuracy for linear classification, which shows that VICReg is a very stable algorithm. Lack of time has prevented us from doing the same for the semi-supervised classification experiments, and the experiments of section 5.2 and 6, but we expect similar conclusion to hold. We compare in Table 1 our results on both tasks against other methods on the validation set of ImageNet. The performance of VICReg is on par with the state of the art without using the negative pairs of SimCLR, the clusters of SwAV, the bag-of-words representations of OBoW, or any asymmetric networks architectural tricks such as the momentum encoder of BYOL and the stop-gradient operation of SimSiam. The performance is comparable to that of Barlow Twins, which shows that VICReg's more explicit way of constraining the variance and comparing views has the same power than maximizing cross-correlations between pairs of twin dimensions. The main advantage of VICReg is the modularity of its objective function and the applicability to multi-modal setups.

## 5.2 TRANSFER TO OTHER DOWNSTREAM TASKS

Following the setup from Misra & Maaten (2020), we train a linear classifier on top of the frozen representations learnt by our pretrained ResNet-50 backbone on a variety of different datasets: the Places205 Zhou et al. (2014) scene classification dataset, the VOC07 Everingham et al. (2010) multi-label image classification dataset and the iNaturalist2018 Horn et al. (2018) fine-grained image classification dataset. We then evaluate the quality of the representations by transferring to other vision tasks including VOC07+12 Everingham et al. (2010) object detection using Faster R-CNN Ren et al. (2015) with a R50-C4 backbone, and COCO Lin et al. (2014) instance segmentation using Mask-R-CNN He et al. (2017) with a R50-FPN backbone. We report the performance in Table 2,

Table 2: **Transfer learning on downstream tasks.** Evaluation of the representations from a ResNet-50 backbone pretrained with VICReg on: (1) linear classification tasks on top of frozen representations, we report Top-1 accuracy (in %) for Places205 Zhou et al. (2014) and iNat18 Horn et al. (2018), and mAP for VOC07 Everingham et al. (2010); (2) object detection with fine-tunning, we report $AP_{50}$ for VOC07+12 using Faster R-CNN with C4 backbone Ren et al. (2015); (3) object detection and instance segmentation, we report AP for COCO Lin et al. (2014) using Mask R-CNN with FPN backbone He et al. (2017). We use † to denote the experiments run by us. Top-3 best self-supervised methods are underlined.

| Method | Linear Classification | | | Object Detection | | |
| --- | --- | --- | --- | --- | --- | --- |
| | Places205 | VOC07 | iNat18 | VOC07+12 | COCO det | COCO seg |
| Supervised | 53.2 | 87.5 | 46.7 | 81.3 | 39.0 | 35.4 |
| MoCo He et al. (2020) | 46.9 | 79.8 | 31.5 | - | - | - |
| PIRL Misra & Maaten (2020) | 49.8 | 81.1 | 34.1 | - | - | - |
| SimCLR Chen et al. (2020a) | 52.5 | 85.5 | 37.2 | - | - | - |
| MoCo v2 Chen et al. (2020c) | 51.8 | 86.4 | 38.6 | 82.5 | 39.8 | 36.1 |
| SimSiam Chen & He (2020) | - | - | - | 82.4 | - | - |
| BYOL Grill et al. (2020) | 54.0 | 86.6 | 47.6 | - | 40.4† | 37.0† |
| SwAV (m-c) Caron et al. (2020) | 56.7 | 88.9 | 48.6 | 82.6 | 41.6 | 37.8 |
| OBoW Gidaris et al. (2021) | 56.8 | 89.3 | - | 82.9 | - | - |
| Barlow Twins Grill et al. (2020) | 54.1 | 86.2 | 46.5 | 82.6 | 40.0† | 36.7† |
| VICReg (ours) | 54.3 | 86.6 | 47.0 | 82.4 | 39.4 | 36.4 |

Table 3: **Evaluation on MS-COCO 5K retrieval tasks.** Comparison of VICReg with the contrastive loss of VSE++ Faghri et al. (2018), and with Barlow Twins, pretrain on the training set of MS-COCO. In all settings, the encoder for text is a word embedding followed by a GRU layer, the encoder for images is a ResNet-152.

| Method | Image-to-text | | | Text-to-Image | | |
| --- | --- | --- | --- | --- | --- | --- |
| | R@1 | R@5 | R@10 | R@1 | R@5 | R@10 |
| Contrastive (VSE++) | 30.3 | 59.4 | 72.4 | 41.3 | 71.1 | 81.2 |
| Barlow Twins | 31.4 | 60,4 | 75.1 | 42.9 | 74.0 | 83.5 |
| VICReg | 33.6 | 62.7 | 77.9 | 45.2 | 76.1 | 84.2 |

VICReg performs on par with most concurrent methods, and better than Barlow Twins, across all classification tasks, but is slightly behind the top-3 on detection tasks.

## 5.3 MULTI-MODAL PRETRAINING ON MS-COCO

One fundamental difference of VICReg compared to Barlow Twins is the way the branches are regularized. In VICReg, both branches are regularized independently, as the covariance term is applied on each branch separately, which works better in the scenarios where the branches are completely different, have different types of architecture and process different types of data. Indeed, the statistics of the output of the two branches can be very different, and the amount of regularization required for each may vary a lot. In Barlow Twins, the regularization is applied on the cross-correlation matrix, which favors the scenarios where the branches produce outputs with similar statistics. We demonstrate the capabilities of VICReg in a multi-modal experiment where we pretrain on pairs of images and corresponding captions on the MS-COCO dataset. We regularize each branch with a different coefficient, which is not possible with Barlow Twins, and we show that VICReg outperforms Barlow Twins on image and text retrieval downstream tasks. Table 3 reports the performance of VICReg against the contrastive loss proposed by VSE++ Faghri et al. (2018), and against Barlow Twins, in the identical setting proposed in Faghri et al. (2018). VICReg outperforms the two by a significant margin.

Table 4: **Effect of incorporating variance and covariance regularization in different methods.** Top-1 ImageNet accuracy with the linear evaluation protocol after 100 pretraining epochs. For all methods, pretraining follows the architecture, the optimization and the data augmentation protocol of the original method using our reimplementation. ME: Momentum Encoder. SG: stop-gradient. PR: predictor. BN: Batch normalization layers after input and inner linear layers in the expander. No Reg: No additional regularization. Var Reg: Variance regularization. Var/Cov Reg: Variance and Covariance regularization. Unmodified original setups are marked by a †.

| Method | ME | SG | PR | BN | No Reg | Var Reg | Var/Cov Reg |
|--------|----|----|----|----|--------|---------|-------------|
| BYOL | ✓ | ✓ | ✓ | ✓ | $69.3^\dagger$ | 70.2 | 69.5 |
| SimSiam | | ✓ | ✓ | ✓ | $67.9^\dagger$ | 68.1 | 67.6 |
| SimSiam | | ✓ | ✓ | | 35.1 | 67.3 | 67.1 |
| SimSiam | | ✓ | | | collapse | 56.8 | 66.1 |
| VICReg | | | ✓ | | collapse | 56.2 | 67.3 |
| VICReg | | | ✓ | ✓ | collapse | 57.1 | 68.7 |
| VICReg | | | | ✓ | collapse | 57.5 | $68.6^\dagger$ |
| VICReg | | | | | collapse | 56.5 | 67.4 |

## 6 ANALYSIS

In this section we study how the different components of our method contribute to its performance, as well as how they interact with components from other self-supervised methods. We also evaluate different scenarios where the branches have different weights and architecture. All reported results are obtained on the linear evaluation protocol, using a ResNet-50 backbone if not mentioned otherwise, and 100 epochs of pretraining, which gives results consistent with those obtained with 1000 epochs of pretraining. The optimization setting used for each experiment is described in Appendix C.

**Asymmetric networks.** We study the impact of different components used in asymmetric architectures and the effects of adding variance and covariance regularization, in terms of performance and training stability. Starting from a simple symmetric architecture with an encoder and an expander without batch normalization, which correspond to VICReg without batch normalization in the expander, we progressively add batch normalization in the inner layers of the expander, a predictor, a stop-gradient operation and a momentum encoder. We use the training protocol and architecture of SimSiam Chen & He (2020) when a stop-gradient is used and the training protocol and architecture of BYOL Grill et al. (2020) when a momentum encoder is used. The predictor as used in SimSiam and BYOL is a learnable module $g_\psi$ that predicts the embedding of a view given the embedding of the other view of the same image. If $z$ and $z'$ are the embeddings of two views of an image, then $p = g_\psi(z)$ and $p' = g_\psi(z')$ are the predictions of each view. The invariance loss function of Eq. (5) is now computed between a batch of embeddings $Z = [z_1, \ldots, z_n]$ and the corresponding batch of predictions $P = [p'_1, \ldots, p'_n]$, then symmetrized:

$$s(Z, Z', P, P') = \frac{1}{2n} \sum_i D(z_i - p'_i) + \frac{1}{2n} \sum_i D(z'_i - p_i), \tag{8}$$

where $D$ is a distance function that depends on the method used. BYOL uses the mean square error between $l_2$-normalized vectors, SimSiam uses the negative cosine similarity loss and VICReg uses the mean square error without $l_2$-normalization. The variance and covariance terms are regularizing the output $Z$ and $Z'$ of the expander, which we empirically found to work better than regularizing the output of the predictor. We compare different settings in Table 4, based on the default data augmentation, optimization and architecture settings of the original BYOL, SimSiam and VICReg methods. In all settings, the absence of BN indicates that BN is also removed in the predictor when one is used.

We analyse first the impact of variance regularization (VR) in the different settings. When using VR, adding a predictor (PR) to VICReg does not lead to a significant change of the performance, which indicates that PR is redundant with VR. In comparison, without VR, the representations collapse, and both stop-gradient (SG) and PR are necessary. Batch normalization in the inner layers of the expander (BN) in VICReg leads to a 1.0% increase in the performance, which is not a big improvement considering that SG and PR without BN is performing very poorly at 35.1%.

Table 5: **Impact of sharing weights or not between branches.** Top-1 accuracy on linear classification with 100 pretraining epochs. The encoder and expander of both branches can share the same architecture and share their weights (SW), share the same architecture with different weights (DW), or have different architectures (DA). The encoders can be ResNet-50, ResNet-101 or ViT-S.

|              | SW R50 | DW R50 | DA R50/R101 | DA R50/ViT-S |
|--------------|--------|--------|-------------|--------------|
| BYOL         | 69.3   | ✗      | ✗           | ✗            |
| SimCLR       | 64.4   | 63.1   | 63.9        | 63.5         |
| Barlow Twins | 68.7   | 64.2   | 65.3        | 63.9         |
| VICReg       | 68.6   | 66.5   | 68.1        | 66.2         |

Finally, incorporating VR with SG or ME further improves the performance by small margins of respectively 0.2% and 0.9%, which might be explained by the fact that these architectural tricks that prevent collapse are not perfectly maintaining the variance of the representations, i.e. very slow collapse is happening with these methods. We explain this intuition by studying the evolution of the standard deviation of the representations during pretraining for BYOL and SimSiam in Appendix D. We then analyse the impact of adding additional covariance regularization (CR) in the different settings, along with variance regularization. We found that optimization with SG and CR is hard, even if our analysis of the average correlation coefficient of the representations during pretraining in Appendix D shows that both fulfill the same objective.

The performance of BYOL and SimSiam slightly drops compared to VR only, except when PR is removed, where SG becomes useless. BN is still useful and improves the performance by 1.3%. Finally with CR, PR does not harm the performance and even improves it by a very small margin. VICReg+PR with 1000 epochs of pretraining exactly matches the score of VICReg (73.2% on linear classification).

**Weight sharing.** Contrary to most self-supervised learning approaches based on Siamese architectures, VICReg has several unique properties: (1) weights do not need to be shared between the branches, each branch's weights are updated independently of the other branch's weights; (2) the branches are regularized independently, the variance and covariance terms are computed on each branch individually; (3) no predictor is necessary unlike with methods where one branch predicts outputs of the other branch. We compare the robustness of VICReg against other methods in different scenarios where the weights of the branches can be shared (SW), not shared (DW), and where the encoders can have different architectures (DA). Among other self-supervised methods, SimCLR and Barlow Twins are the only ones that can handle these scenarios. The asymmetric methods that are based on a discrepancy between the branches requires either the architecture or the weights to be shared between the branches. The performance drops by 2.1% with VICReg and 4.5% with Barlow Twins, between the shared weights scenario (SW) and the different weight scenario (DW). The difference between VICReg and Barlow Twins is also significant in scenarios with different architectures, in particular VICReg performs better than Barlow Twins by 2.8% with ResNet-50/ResNet-101 and better by 2.3% with ResNet-50/ViT-S Dosovitskiy et al. (2021). This shows that VICReg is more robust than Barlow Twins in these kind of scenarios. The performance of SimCLR remains stable across scenarios, but is significantly worse than the performance of VICReg. Importantly, **the ability of VICReg to function with different parameters, architectures, and input modalities for the branches widens the applicability to joint-embedding SSL to many applications, including multi-modal signals.**

## 7 CONCLUSION

We introduced VICReg, a simple approach to self-supervised learning based on a triple objective: learning invariance to different views with a invariance term, avoiding collapse of the representations with a variance preservation term, and maximizing the information content of the representation with a covariance regularization term. VICReg achieves results on par with the state of the art on many downstream tasks, but is not subject to the same limitations as most other methods, particularly because it does not require the embedding branches to be identical or even similar.

**Acknowledgement.** Jean Ponce was supported in part by the French government under management of Agence Nationale de la Recherche as part of the "Investissements d'avenir" program, reference ANR-19-P3IA-0001 (PRAIRIE 3IA Institute), the Louis Vuitton/ENS Chair in Artificial Intelligence and the Inria/NYU collaboration. Adrien Bardes was supported in part by a FAIR/Prairie CIFRE PhD Fellowship. The authors wish to thank Jure Zbontar for the BYOL implementation, Stéphane Deny for useful comments on the paper, and Li Jing, Yubei Chen, Mikael Henaff, Pascal Vincent and Geoffrey Zweig for useful discussions. We thank Quentin Duval and the VISSL team for help obtaining the results of table 2.

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

## A ALGORITHM

**Algorithm 1:** VICReg pytorch pseudocode.

```
# f: encoder network, lambda, mu, nu: coefficients of the
    invariance, variance and covariance losses, N: batch size
    , D: dimension of the representations
# mse_loss: Mean square error loss function, off_diagonal:
    off-diagonal elements of a matrix, relu: ReLU activation
    function

for x in loader: # load a batch with N samples
    # two randomly augmented versions of x
    x_a, x_b = augment(x)

    # compute representations
    z_a = f(x_a) # N x D
    z_b = f(x_b) # N x D

    # invariance loss
    sim_loss = mse_loss(z_a, z_b)

    # variance loss
    std_z_a = torch.sqrt(z_a.var(dim=0) + 1e-04)
    std_z_b = torch.sqrt(z_b.var(dim=0) + 1e-04)
    std_loss = torch.mean(relu(1 - std_z_a)) + torch.mean(
        relu(1 - std_z_b))

    # covariance loss
    z_a = z_a - z_a.mean(dim=0)
    z_b = z_b - z_b.mean(dim=0)
    cov_z_a = (z_a.T @ z_a) / (N - 1)
    cov_z_b = (z_b.T @ z_b) / (N - 1)
    cov_loss = off_diagonal(cov_z_a).pow_(2).sum() / D
              + off_diagonal(cov_z_b).pow_(2).sum() / D

    # loss
    loss = lambda * sim_loss + mu * std_loss + nu * cov_loss

    # optimization step
    loss.backward()
    optimizer.step()
```

## B  RELATION TO OTHER SELF-SUPERVISED METHODS

We compare here VICReg with other methods in terms of methodology, and we discuss the mechanisms used by these methods to avoid collapse and to learn representations, and how they relate to VICReg. We synthesize and illustrate the differences between these methods in Figure 2.

**Relation to Barlow Twins** Zbontar et al. (2021). VICReg uses the same decorrelation mechanism as Barlow Twins, which consists in penalizing the off-diagonal terms of a covariance matrix computed on the embeddings. However, Barlow Twins uses the cross-correlation matrix where each entry in the matrix is a cross-correlation between two vectors $z^i$ and $z'^j$, from the two branches of the siamese architecture. Instead of using cross-correlations, we simply use the covariance matrix of each branch individually, and the variance term of VICReg allows us to get rid of standardization. Indeed, Barlow Twins forces the correlations between pairs of vectors $z^i$ and $z'^i$ from the same dimension $i$ to be 1. Without normalization, this target value of 1 becomes arbitrary and the vectors take values in a wider range. Moreover, there is an undesirable phenomenon happening in Barlow Twins, the embeddings before standardization can shrink and become constant to numerical precision, which could cause numerical instabilities. In practice, this is solved by adding a constant scalar in the denominator of standardization of the embeddings. Without normalization, VICReg naturally avoids this edge case.

**Relation to W-MSE** Ermolov et al. (2021). The whitening operation of W-MSE consists in computing the inverse covariance matrix of the embeddings and use its square root as a whitening operator on the embeddings. Using this operator has two downsides. First, matrix inversion is a very costly and potentially unstable operation. VICReg does not need to inverse the covariance matrix. Second, as mentioned in Ermolov et al. (2021) the whitening operator is constructed over several consecutive iteration batches and therefore might have a high variance, which biases the estimation of the mean-squared error. This issue is overcome in practice by a batch slicing strategy, where the whitening operator is computed over randomly constructed sub-batches. VICReg does not apply any operator on the embeddings, but instead regularizes the variance and covariance of the embeddings using an additional constraint.

**Relation to BYOL and SimSiam** Grill et al. (2020); Chen & He (2020). The core components that avoid collapse in BYOL and SimSiam are the average moving weights and the stop-gradient operation on one side of their asymmetric architecture, which play the role of the repulsive term used in other methods. Our experiments in Appendix D.8 show that in addition to preventing collapse, these components also have a decorrelation effect. In addition, we have conducted the following experiment: We compute the correlation matrix of the final representations obtained with SimSiam, BYOL, VICReg and VICReg without covariance regularization. We measure the average correlation coefficient and observe that this coefficient is much smaller for SimSiam, BYOL and VICReg, compared to VICReg without covariance regularization. We observe in Figure 5 that even without covariance regularization, SimSiam and BYOL naturally minimize the average correlation coefficient of the representations. VICReg replaces the moving average weights and the stop-gradient operation, which are architectural trick that require some dependency between the branches, by an explicit constraint on the variance and the covariance of both embeddings separately, which achieves the same goal of decorrelating the representations and avoiding collapse, while being clearer, more interpretable, and working with independent branches.

**Relation to SimCLR, SwAV and OBoW** Caron et al. (2020); Chen et al. (2020a); Gidaris et al. (2021). Contrastive and clustering based self-supervised algorithms rely on direct comparisons between elements of negative pairs. In the case of SimCLR, the negative pairs involve embeddings mined from the current batch, and large batch sizes are required. Despite the fact that SwAV computes clusters using elements in the current batch, it does not seem to have the same dependency on batch size. However, it still requires a lot of prototype vectors for negative comparisons between embeddings and codes. VICReg eliminates the negative comparisons and replace them by an explicit constraint on the variance of the embeddings, which efficiently plays the role of a negative term between the vectors. SwAV can also be interpreted as a distillation method, where a teacher network produces quantized vectors, used as target for a student network. Ensuring an equal partition of the quantized vectors in different bins or clusters effectively prevents collapse. OBOW can also be interpreted under the same framework. The embeddings are bag-of-words over a vocabulary of visual features, and collapse is avoided by the underlying quantization operation.

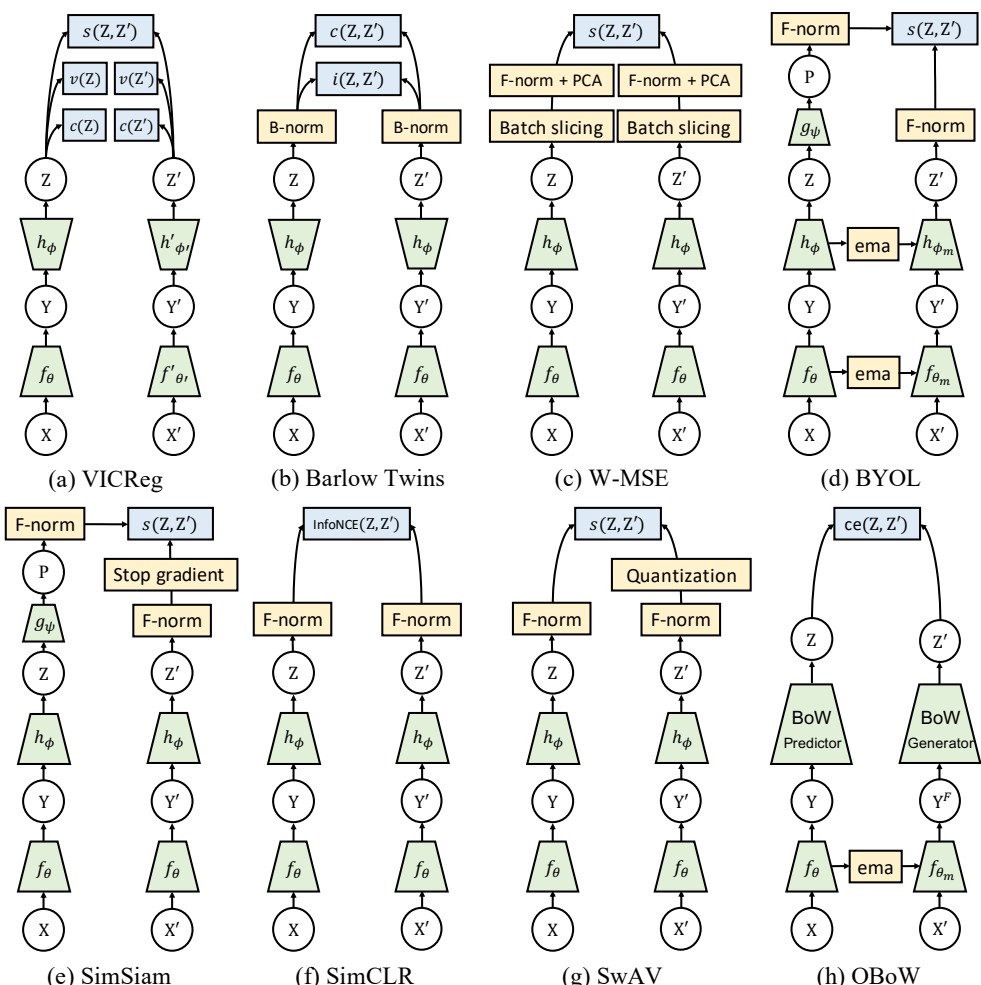

Figure 2: **Conceptual comparison between different self-supervised methods.** The inputs $X$ and $X'$ are fed to an encoder $f$ with weights $\theta$. The representations $Y$ and $Y'$ are further processed by a network $h$ with weights $\psi$. $h$ can be a projector (narrowing trapeze) that reduces the dimensionality of the representations, or an expander (widening trapeze) that increases their dimensionality. A criterion is finally applied on the embeddings $Z$ and $Z'$. VICReg (a) works when both branches have encoders $f$ and $f'$ with different architectures and sets of weights $\theta$ and $\theta'$. Each branch's variance and covariance are regularized by regularizers $v$ and $c$, and the distance between both branches is minimized with a mean-squared error loss $s$. Barlow Twins (b) uses a loss $c$ to decorrelate pairs of different dimensions in the batch-wise normalized (B-Norm) embeddings, and learns invariance with a loss $i$ that makes similar dimensions highly correlated. W-MSE (c) uses a batch slicing operation that shuffles batches into small sub-batches, and apply PCA as a whitening operation on the feature-wise normalized (F-Norm) embeddings of each sub-batch. BYOL (d) has an asymmetric architecture where the weights $\theta_m$ of one encoder are an exponential moving average (ema) of the other encoder's weights $\theta$. A predictor $g$ with weights $\psi$ is used in the branch with learnable weights. SimSiam (e) uses a predictor on one branch and a stop-gradient operation (sg) on the other one. SimCLR (f) uses the InfoNCE contrastive loss where all the feature-wise normalized embeddings are compared between them inside a batch. Samples from distorted versions of the same input are brought close to each other, while other samples are pushed away. SwAV (g) quantizes the feature-wise normalized embeddings of a branch and use it as target for the other one. OBoW (h) uses bag-of-words (BoW) representations and a cross-entropy loss to compare the BoW generated by a teacher network from the feature maps $Y^F$ of the encoder, to the BoW predicted by a student network. Green blocks: parametric functions; yellow boxes: non-parametric functions; blue boxes: objective functions.

## C  ADDITIONAL IMPLEMENTATION DETAILS

### C.1  DATA AUGMENTATION

We follow the image augmentation protocol first introduced in SimCLR Chen et al. (2020a) and now commonly used by similar approaches based on siamese networks Caron et al. (2020); Grill et al. (2020); Chen & He (2020); Zbontar et al. (2021). Two random crops from the input image are sampled and resized to $224 \times 224$, followed by random horizontal flip, color jittering of brightness, contrast, saturation and hue, Gaussian blur and random grayscale. Each crop is normalized in each color channel using the ImageNet mean and standard deviation pixel values. In more details, the exact set of augmentations is based on BYOL Grill et al. (2020) data augmentation pipeline but is symmetrised. The following operations are performed sequentially to produce each view:

- Random cropping with an area uniformly sampled with size ratio between 0.08 to 1.0, followed by resizing to size $224 \times 224$. `RandomResizedCrop(224, scale=(0.08, 0.1))` in PyTorch.
- Random horizontal flip with probability 0.5.
- Color jittering of brightness, contrast, saturation and hue, with probability 0.8. `ColorJitter(0.4, 0.4, 0.2, 0.1)` in PyTorch.
- Grayscale with probability 0.2.
- Gaussian blur with probability 0.5 and kernel size 23.
- Solarization with probability 0.1.
- color normalization with mean (0.485, 0.456, 0.406) and standard deviation (0.229, 0.224, 0.225).

### C.2  IMAGENET EVALUATION

**Linear evaluation.** We follow standard procedure and train a linear classifier on top of the frozen representations of a ResNet-50 pretrained with VICReg. We use the SGD optimizer with a learning rate of 0.02, a weight decay of $10^{-6}$, a batch size of 256, and train for 100 epochs. The learning rate follows a cosine decay. The training data augmentation pipeline is composed of random cropping and resize of ratio 0.2 to 1.0 with size $224 \times 224$, and random horizontal flips. During evaluation the validation images are simply center cropped and resized to $224 \times 224$.

**Semi-supervised evaluation.** We train a linear classifier and fine-tune the representations using 1 and 10% of the labels. We use the SGD optimizer with no weight decay and a batch size of 256, and train for 20 epochs. We perform a grid search on the values of the encoder and linear head learning rates. In the 10% of labels case, we use a learning rate of 0.01 for the encoder and 0.1 for the linear head. In the 1% of labels case we use 0.03 for the encoder and 0.08 for the linear head. The two learning rates follow a cosine decay schedule. The training data and validation augmentation pipelines are identical to the linear evaluation data augmentation pipelines.

### C.3  TRANSFER LEARNING

We use the VISSL library Goyal et al. (2021) for linear classification tasks and the detectron2 library Wu et al. (2019) for object detection and segmentation tasks.

**Linear classification.** We follow standard protocols Misra & Maaten (2020); Caron et al. (2020); Zbontar et al. (2021) and train linear models on top of the frozen representations. For VOC07 Everingham et al. (2010), we train a linear SVM with LIBLINEAR Fan et al. (2008). The images are center cropped and resized to $224 \times 224$, and the C values are computed with cross-validation. For Places205 Zhou et al. (2014) we use SGD with a learning rate of 0.003, a weight decay of 0.0001, a momentum of 0.9 and a batch size of 256, for 28 epochs. The learning rate is divided by 10 at epochs 4, 8 and 12. For Inaturalist2018 Horn et al. (2018), we use SGD with a learning rate of 0.005, a weight decay of 0.0001, a momentum of 0.9 and a batch size of 256, for 84 epochs. The learning rate is divided by 10 at epochs 24, 48 and 72.

**Object detection and instance segmentation.** Following the setup of He et al. (2020); Zbontar et al. (2021), we use the `trainval` split of VOC07+12 with 16K images for training and a Faster

R-CNN C-4 backbone for 24K iterations with a batch size of 16. The backbone is initialized with our pretrained ResNet-50 backbone. We use a learning rate of 0.1, divided by 10 at iteration 18K and 22K, a linear warmup with slope of 0.333 for 1000 iterations, and a region proposal network loss weight of 0.2. For COCO we use Mask R-CNN FPN backbone for 90K iterations with a batch size of 16, a learning rate of 0.04, divided by 10 at iteration 60K and 80K and with 50 warmup iterations.

## C.4 ANALYSIS

We give here implementation details on the results of Table 4 with BYOL and SimSiam, as well as the default setup for VICReg with 100 epochs of pretraining, used in all our ablations included in Appendix D. For both BYOL and SimSiam experiments, the variance criterion has coefficient $\mu = 1$ and the covariance criterion has coefficient $\nu = 0.01$, the data augmentation pipeline and the architectures of the expander and predictor exactly follow the pipeline and architectures described in their paper. The linear evaluation setup of each methods follows closely the setup described in the original papers.

**BYOL setup.** We use our own BYOL implementation in PyTorch, which outperforms the original implementation for 100 epochs of pretraining (69.3% accuracy on the linear evaluation protocol against 66.5% for the original implementation) and matches its performance for 1000 epochs of pretraining. We use the LARS optimizer You et al. (2017), with a learning rate of $base\_lr *$ $batch\_size/256$ where $base\_lr = 0.45$, and $batch\_size = 4096$, a weight decay of $10^{-6}$, an eta value of 0.001 and a momentum of 0.9, for 100 epoch of pretraining with 10 epochs of warmup. The learning rate follows a cosine decay schedule. The initial value of the exponential moving average factor is 0.99 and follows a cosine decay schedule.

**SimSiam setup.** We use our own implementation of SimSiam, which reproduces exactly the performance reported in the paper Chen & He (2020). We use SGD with a learning rate of $base\_lr * batch\_size/256$ where $base\_lr = 0.05$, $batch\_size = 2048$, with a weight decay of 0.0001 and a momentum of 0.9 for 100 epochs of pretraining and 10 epochs of warmup. The learning rate of the encoder and the expander follow a cosine decay schedule while the learning rate of the predictor is kept fixed.

**VICReg setup.** The setting of VICReg's experiments is identical to the setting described in section 4.2, except that the number of pretraining epochs is 100 and the base learning rate is 0.3. The base learning rates used for the batch size study are 0.8, 0.5 and 0.4 for batch size 128, 256 and 512 respectively, and 0.3 for all other batch sizes. When a predictor is used, it has a similar architecture as the expander described in section 4.2, but with 2 layers instead of 3, which gives better results in practice.

## D ADDITIONAL RESULTS

### D.1 OTHER RESNET ARCHITECTURES

Table 9 reports the performance of VICReg on linear classification with large ResNet architectures. We focus on the wider family of ResNet Zagoruyko & Komodakis (2016) and aggregated ResNet Xie et al. (2017), and we consider two ways of widening a standard ResNet. First, we follow standard practice in recent self-supervised learning work Caron et al. (2020); Grill et al. (2020); Chen et al. (2020a) and multiple by 2 or 4 the number of filters in every convolutional layer, which also has the effect of multiplying the dimensionality of the representations. Second, as originally proposed in Zagoruyko & Komodakis (2016), we only multiply the number of filters in the bottleneck layers, which does not increases the dimensionality of the representations. We call this architecture Narrow ResNet (with prefix N- in Table 9). The main observation we make is the dependency of VICReg on the dimensionality of the representation. Using the narrow architecture, the performance of VICReg, jumps from 73.2% top-1 accuracy on linear classification with a ResNet-50, to 74.7% with Narrow ResNet-50 (x2), which is a 1.5% improvement and 76.0% with Narrow ResNet-50 (x4), which is a 2.8% improvement. We observe a similar trend going from ResNet-50 to ResNet-50 (x2), which is a 2.3% improvement but the performance completely saturates with ResNet-50 (x4), which is a 0.1% improvement over ResNet-50 (x2). Table 10 reports the performance of VICReg on semi-supervised classification with large ResNet architectures. VICReg combined with a ResNet-50 (x2) outperforms the current state-of-the-art methods BYOL and SimCLR, using this encoder architecture. Our largest

Table 6: **Evaluation on ESC-50.** Evaluation of the representations obtained with a ResNet-18 backbone pretrained with VICReg on ESC-50 Piczak (2015) by processing jointly a raw audio time-series and its corresponding time-frequency representation. The supervised baseline corresponds to a ResNet-18 trained on the time-frequency representation in a supervised way. We report Top-1 accuracy on the validation set (in %).

| Method | Top-1 |
|---|---|
| Supervised baseline | 72.7 |
| Barlow Twins | 75.4 |
| VICReg | 78.4 |

model ResNet-200 (x2) performs lower than BYOL when 1% of the labels are used but is on par with 10% of the labels. These results demonstrate the capabilities of VICReg to scale up when large architectures are used.

### D.2 Pretraining and evaluation on ESC-50 audio classification

We demonstrate the ability of VICReg to function in a setting where the branches have different architectures by pretraining on the ESC-50 audio dataset Piczak (2015), which is an environmental sound classification dataset with 50 classes. We jointly embedded a raw audio time-series representation on one branch, with its corresponding time-frequency representation on the other branch. We use the standard split of ESC-50 Piczak (2015), composed of 1600 training audio samples and 400 validation sample. The raw audio encoder is a 1-dimensional ResNet-18 with output dimension 384. The time-frequency image representation is the mel spectrogram with 1 channel of the raw audio, that we normalize between 0 and 1, and that is processed be a ResNet-18 with output dimension of 512. We use the AdamW optimizer with learning rate 0.0005 for 100 epochs of pretraining.

Table 6 reports the performance of a linear classifier trained one the frozen representations obtained with VICReg and Barlow Twins to a simple supervised baseline where we train a ResNet-18 on the time-frequency representation in a supervised way. VICReg performs better by 5.7% than our supervised baseline, and better by 3.0% than Barlow Twins. We give more details in Appendix **??**. Current best approaches that report around 95% accuracy on this task uses tricks such as heavy data augmentation or pretraining on larger audio and video datasets. With this experiment, our purpose is not to push the state of the art on ESC-50, but merely to demonstrate the applicability of VICReg to settings with multiple architectures and input modalities.

### D.3 K-nearest-neighbors

Following recent protocols Caron et al. (2020); Wu et al. (2018); Zhuang et al. (2019), we evaluate the learnt representations using K-nearest-neighbors classifiers built on the training set of ImageNet and evaluated on the validation set of ImageNet. We report the results with K=20 and K=200 in Table 11. VICReg performs slightly lower than other methods in the 20-NN case but remains competitive in the 200-NN case. These results with K-NN classifiers demonstrate the potential applicability of VICReg to downstream tasks based on nearest neighbors search, such as content retrieval in images or videos.

### D.4 Loss function coefficients.

Table 7 reports the performance for various values of the loss term coefficients in Eq. (6). Without variance regularization the representations immediately collapse to a single vector and the covariance term, which has no repulsive effect preventing collapse, has no impact. The invariance term is absolutely necessary and without it the network can not learn any good representations. By simply using the invariance term and variance regularization, which is a very simple baseline, VICReg still reaches an accuracy of $57.5\%$. These results show that variance and covariance regularizations have complementary effects, and that both are required.

On ImageNet, we choose the final coefficients the following way. First, we have empirically found that using very different values for $\lambda$ and $\mu$, or taking $\lambda = \mu$ with $\nu > \mu$ leads to unstable training. On the other hand taking $\lambda = \mu$ and picking $\nu < \mu$ leads to stable convergence, with the exact value picked for $mu$ having very limited influence on the final linear classification accuracy. We have found

Table 7: **Impact of variance-covariance regularization.** Inv: a invariance loss is used, $\lambda > 0$, Var: variance regularization, $\mu > 0$, Cov: covariance regularization, $\nu > 0$, in Eq. (6).

| Method | $\lambda$ | $\mu$ | $\nu$ | Top-1 |
|---|---|---|---|---|
| Inv | 1 | 0 | 0 | collapse |
| Inv + Cov | 25 | 0 | 1 | collapse |
| Inv + Cov | 0 | 25 | 1 | collapse |
| Inv + Var | 1 | 1 | 0 | 57.5 |
| Inv + Var + Cov (VICReg) | 1 | 1 | 1 | collapse |
| | 1 | 10 | 1 | collapse |
| | 10 | 1 | 1 | collapse |
| | 5 | 5 | 1 | 68.1 |
| | 10 | 10 | 1 | 68.2 |
| | 25 | 25 | 1 | 68.6 |
| | 50 | 50 | 1 | 68.3 |

Table 8: **Impact of normalization.** Std: variables are centered and divided by their standard deviation over the batch. This is applied or not to the embedding and the expander hidden layers. $l_2$: the embedding vectors are $l_2$-normalized.

| Representation | Embedding | Top-1 |
|---|---|---|
| Std | None | 68.6 |
| Std | Std | 68.4 |
| None | Std | 67.4 |
| Std | None | 67.2 |
| None | $l_2$ | 65.1 |

that setting $lambda = mu = 25$ and $nu = 1$ works best (by a small margin) for Imagenet but we have also obtained excellent results on MNIST and Cifar-10 and 100 using these exact same values. We could easily have tuned these parameters by cross-validation on the validation sets of these two smaller datasets.

### D.5 NORMALIZATIONS

VICReg is the first self-supervised method for joint-embedding architectures we are aware of that does not require normalization. Contrary to SimSiam, W-MSE, SwAV and BYOL, and others, the embedding vectors are not projected on the unit sphere. Contrary to Barlow Twins, they are not standardized (equivalent to batch normalization without the adaptive parameters). Table 8 shows that the best settings do not involve any normalization of the embeddings, whether it is batch-wise or feature-wise (as in $l_2$ normalization). Whenever the embeddings are standardized (lines 3 and 5 in the table) the covariance matrix of Eq. (3) becomes the normalized auto-correlation matrix with coefficients between -1 and 1. This hurts the accuracy by 0.2%. We observe that when unconstrained, the coefficients in the covariance matrix take values in a wider range, which seems to facilitate the training process. Standardization is still an important component that helps stabilize the training when used in the hidden layers of the expander, and the performance drops by 1.2% when it is removed. Projecting the embeddings on the unit sphere implicitly constrains their standard deviation along the batch dimension to be $1/\sqrt{d}$, where $d$ is the dimension of the vectors. We change the invariance term of Eq. (5) to be the mean square error between $l_2$-normalized vectors, and the target $\gamma$ in the variance term of Eq. (1) is set to $1/\sqrt{d}$ instead of 1, forcing the standard deviation to get closer to $1/\sqrt{d}$, and the vectors to be spread out on the unit sphere. This puts a lot more constraints on the network and the performance drops by 3.5%.

Table 9: **Linear classification with large architectures.** Top-1 accuracy comparison between different methods using various encoder architectures. For all VICReg results, the output dimensionality of the expander is 8192. N-R stands for Narrow ResNet, where only the bottleneck convolutional layers are widen.

| Method | Arch. | Param. | Repr. | Top-1 | Top-5 |
|--------|-------|--------|-------|-------|-------|
| SimCLR Chen et al. (2020a) | R50 (x2) | 93M | 4096 | 74.2 | 92.0 |
| | R50 (x4) | 375M | 8192 | 76.5 | 93.2 |
| SwAV Caron et al. (2020) | R50 (x2) | 93M | 4096 | 77.3 | - |
| | R50 (x4) | 375M | 8192 | 77.9 | - |
| | R50 (x5) | 586M | 10240 | 78.5 | - |
| BYOL Grill et al. (2020) | R50 (x2) | 93M | 4096 | 77.4 | 93.6 |
| | R50 (x4) | 375M | 8192 | 78.6 | 94.2 |
| | R200 (x2) | 250M | 4096 | 79.6 | 94.8 |
| VICReg (ours) | N-R50 (x2) | 66M | 2048 | 74.7 | 91.9 |
| | N-R50 (x4) | 221M | 2048 | 76.0 | 92.4 |
| | R50 (x2) | 93M | 4096 | 75.5 | 92.1 |
| | R50 (x4) | 375M | 8192 | 75.6 | 92.2 |
| | RNXT101-32-16 | 191M | 2048 | 76.1 | 92.3 |
| | R200 (x2) | 250M | 4096 | 77.3 | 93.3 |

Table 10: **Semi-supervised classification with large architectures.** Top-1 accuracy comparison between different methods using various encoder architectures. For all VICReg results, the output dimensionality of the expander is 8192.

| Method | Arch. | Param. | Repr. | Top-1 | | Top-5 | |
|--------|-------|--------|-------|-------|-------|-------|-------|
| | | | | 1% | 10% | 1% | 10 % |
| SimCLR Chen et al. (2020a) | R50 (x2) | 93M | 4096 | 58.5 | 71.7 | 83.0 | 91.2 |
| | R50 (x4) | 375M | 8192 | 63.0 | 74.4 | 85.8 | 92.6 |
| BYOL Grill et al. (2020) | R50 (x2) | 93M | 4096 | 62.2 | 73.5 | 84.1 | 91.7 |
| | R50 (x4) | 375M | 8192 | 69.1 | 75.7 | 87.9 | 92.5 |
| | R200 (x2) | 250M | 4096 | 71.2 | 77.7 | 89.5 | 93.7 |
| VICReg (ours) | R50 (x2) | 93M | 4096 | 62.6 | 73.9 | 84.5 | 91.8 |
| | R200 (x2) | 250M | 4096 | 68.8 | 77.3 | 88.2 | 93.6 |

Table 11: **K-NN classifiers on ImageNet.** Top-1 accuracy with 20 and 200 nearest neighbors.

| Method | 20-NN | 200-NN |
|--------|-------|--------|
| NPID Wu et al. (2018) | - | 46.5 |
| LA Zhuang et al. (2019) | - | 49.4 |
| PCL Li et al. (2021) | 54.5 | - |
| BYOL Grill et al. (2020) | 66.7 | 64.9 |
| SwAV Caron et al. (2020) | 65.7 | 62.7 |
| Barlow Twins Zbontar et al. (2021) | 64.8 | 62.9 |
| VICReg | 64.5 | 62.8 |

Table 12: **Impact of expander dimensionality.** Top-1 accuracy on the linear evaluation protocol with 100 pretraining epochs.

| Dimensionality | 256 | 512 | 1024 | 2048 | 4096 | 8192 | 16834 |
|---|---|---|---|---|---|---|---|
| Top-1 | 55.9 | 59.2 | 62.4 | 65.1 | 67.3 | 68.6 | 68.8 |

Table 13: **Impact of batch size.** Top-1 accuracy on the linear evaluation protocol with 100 pretraining epochs.

| Batch size | 128 | 256 | 512 | 1024 | 2048 | 4096 |
|---|---|---|---|---|---|---|
| Top-1 | 67.3 | 67.9 | 68.2 | 68.3 | 68.6 | 67.8 |

### D.6 EXPANDER NETWORK ARCHITECTURE

VICReg borrows the decorrelation mechanism of Barlow Twins Zbontar et al. (2021) and we observe that it therefore has the same dependency on the dimensionality of the expander network. Table 12 reports the impact of the width and depth of the expander network. The dimensionality corresponds the number of hidden and output units in the expander network during pretraining. As the dimensionality increases, the performance dramatically increases from 55.9% top-1 accuracy on linear evaluation with a dimensionality of 256, to 68.8% with dimensionality 16384. The performance tends to saturate as the difference between dimensionality 8192 and 16384 is only of 0.2%.

### D.7 BATCH SIZE

Contrastive methods suffer from the need of a lot of negative examples which can translate into the need for very large batch sizes Chen et al. (2020a). Table 13 reports the performance on linear classification when the size of the batch varies between 128 and 4096. For each value of batch size, we perform a grid search on the base learning rate described in Appendix C.4. We observe a $0.7\%$ and $1.2\%$ drop in accuracy with small batch size of 256 and 128 which is comparable with the robustness to batch size of Barlow Twins Zbontar et al. (2021) and SimSiam Chen & He (2020), and a $0.8\%$ drop with a batch size of 4096, which is reasonable and allows our method to be very easily parallelized on multiple GPUs.

### D.8 COMBINATION WITH BYOL AND SIMSIAM

BYOL Grill et al. (2020) and SimSiam Chen & He (2020) rely on a effective but difficult to interpret mechanism for preventing collapse, which may lead to instabilities during the training. We incorporate our variance regularization loss into BYOL and SimSiam and show that it helps stabilize the training and offers a small performance improvement. For both methods, the results are obtained using our own implementation and the exact same data augmentation and optimization settings as in their original paper. The variance and covariance regularization losses are incorporated with a factor of $\mu = 1$ for variance and $\nu = 0.01$ for covariance. We report in Figure 3 the improvement obtained over these methods on the linear evaluation protocol for different number of pre-training epochs. For BYOL the improvement is of 0.9% with 100 epochs and becomes less significant as the number of pre-training epochs increases with a 0.2% improvement with 1000 epochs. This indicates that variance regularization makes BYOL converge faster. In SimSiam the improvement is not as significant. We plot in Figure 4 the evolution of the standard deviation computed along each dimension and averaged across the dimensions of the representation and the embeddings, during BYOL and SimSiam pretraining. For both methods, the standard deviation computed on the embeddings perfectly matches $1/\sqrt{d}$ where $d$ is the dimension of the embeddings, which indicates that the embeddings are perfectly spread-out across the unit sphere. This translates in an increased standard deviation at the representation level, which seems to be correlated to the performance improvement. We finally study in Figure 5 the evolution of the average correlation coefficient, during pretraining of BYOL and SimSiam, with and without variance and covariance regularization. The average correlation coefficient is computed by averaging the off-diagonal coefficients of the

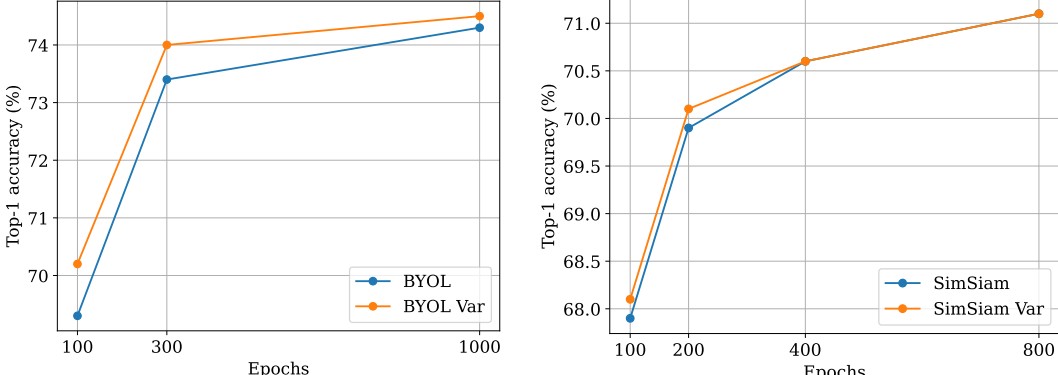

Figure 3: **Incorporating variance regularization in BYOL and SimSiam.** Top-1 accuracy on the linear evaluation protocol for different number of pretraining epochs. For both methods pre-training follows the optimization and data augmentation protocol of their original paper but is based on our implementation. *Var* indicates variance regularization

correlation matrix of the representations:

$$\frac{1}{2d(d-1)} \sum_{i \neq j} C(Y)^2_{i,j} + C(Y')^2_{i,j}, \tag{9}$$

where $Y$ and $Y'$ are the standardized representations and $C$ is defined in Eq. (3). In BYOL this coefficient is much lower using covariance regularization, which translate in a small improvement of the performance, according to Table 4. We do not observe the same improvement in SimSiam, both in terms of correlation coefficient, and in terms of performance on linear classification. The average correlation coefficient is correlated with the performance, which motivates the fact that decorrelation and redundancy reduction are core mechanisms for learning self-supervised representations.

## E    RUNNING TIME

We report in Table 14, the running time of VICReg in comparison with other methods. All methods are run by us on 32 Tesla V100 GPUs. Each method offers a different trade-off between running time, memory and performance. SwAV is a very fast algorithm which use less memory and run faster than the other methods but with a lower performance, multi-crop helps the performance at the cost of additional compute and memory usage. BYOL has the highest memory requirement, which is due to the need of storing the target network weights. Finally, Barlow Twins and VICReg offer an interesting trade-off, consuming less memory than BYOL and SwAV with multi-crop, and running faster than SwAV with multi-crop, but with a slightly worse performance. The difference of 1h running time between Barlow Twins and VICReg is probably due to implementation details not related to the method.

Table 14: **Running time and peak memory.** Comparison between different methods, the training is distributed on 32 Tesla V100 GPUs, the running time is measured over 100 epochs and the peak memory is measured on a single GPU. We report top-1 accuracy (%) on linear classification on top of the frozen representations.

| Method | time / 100 epochs | peak memory / GPU | Top-1 accuracy (%) |
|---|---|---|---|
| SwAV | 9h | 9.5G | 71.8 |
| SwAV (w/ multi-crop) | 13h | 12.9G | 75.3 |
| BYOL | 10h | 14.6G | 74.3 |
| Barlow Twins | 12h | 11.3G | 73.2 |
| VICReg | 11h | 11.3G | 73.2 |

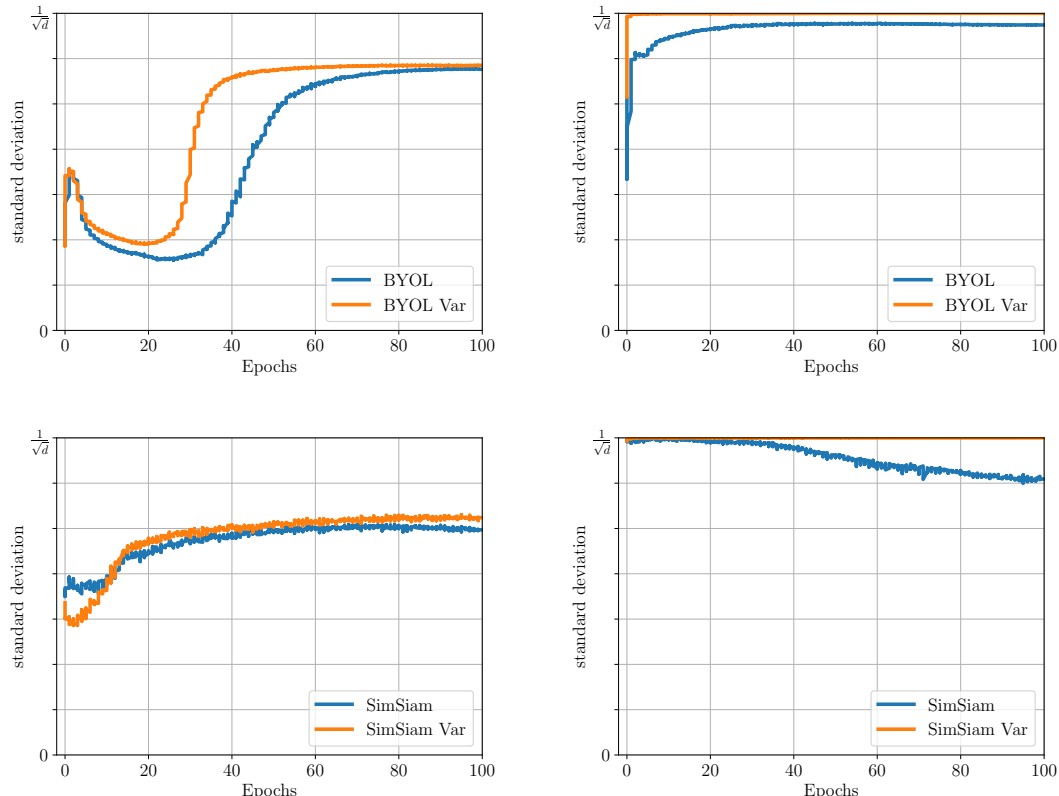

Figure 4: **Standard deviation of the features during BYOL and SimSiam pretraining.** Evolution of the average standard deviation of each dimension of the features with and without variance regularization (Var). left: the standard deviation is measured on the representations, right: the standard deviation is measured on the embeddings.

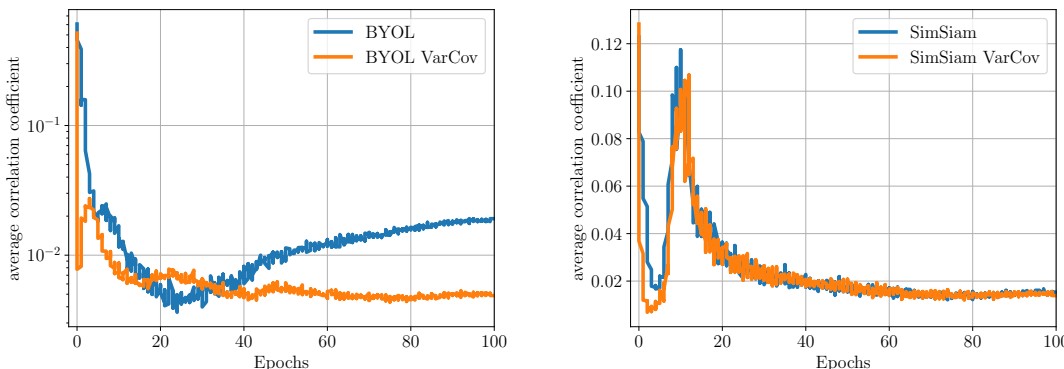

Figure 5: **Average correlation coefficient of the features during BYOL and SimSiam pretraining.** Evolution of the average correlation coefficient measured by averaging the off-diagonal terms of the correlation matrix of the representations with BYOL, BYOL with variance-covariance regularization (BYOL VarCov), SimSiam, and SimSiam with variance-covariance regularization (SimSiam VarCov).

