# OpenReview forum: "VICReg: Variance-Invariance-Covariance Regularization for Self-Supervised Learning"
_ICLR.cc/2022/Conference — ICLR 2022 Poster_

### Official Review · Reviewer_dMLo · 2021-10-26

**Correctness:** 3
**Technical Novelty And Significance:** 2
**Empirical Novelty And Significance:** 2
**Recommendation:** 3
**Confidence:** 4

**Main Review:**

# Strengths

- The overall exposition of the paper is clear and easy to follow.
- The proposed method is simpler than the previously proposed self-supervised learning techniques. It is agreeable that the variance and covariance terms prevent the collapse of representations.
- The ability to handle the heterogeneous encoding networks seems to be a meaningful improvement.
- The proposed method requires a moderately sized batch of 2048.

# Weaknesses

- It is unclear that the collapse of representations, the main problem tackled by the paper, is the major bottleneck in self-supervised learning. The experimental results presented in Table 1 and Table 2 are okay, but not pushing the boundary of self-supervised learning.
- While Table 3 and Table 5 showed that VICReg is more suitable for using heterogeneous encoders, the necessity of heterogeneous encoders is not demonstrated very clearly, because the setting is not practical. The performances reported in Table 3 are far from the state-of-the-art, and in Table 5, the shared weight setting performs best. A more natural setting, such as representation learning for multi-modal data as in VSE [1], should be investigated.
- The contributions of the variance term and the covariance term are not well analyzed. Table 4 is supposed to show the contributions, but it lacks CovReg column so that the conclusion from the table is somewhat vague. Additional efforts for illustrating the effect of the variance and the covariance terms will make the paper more persuasive.
- The difference from Barlow Twins needs to be elaborated in detail. Otherwise, the proposed method is conceived as a minor improvement over Barlow Twins. I found that the definition of the covariance term is meaningfully different from that of Barlow Twins, but it is not emphasized.

[1] Faghri, Fartash, et al. "VSE++: Improving visual-semantic embeddings with hard negatives." arXiv preprint arXiv:1707.05612 (2017).

**Summary Of The Paper:**

The paper proposes a novel objective function for self-supervised representation learning. The objective function consists of three terms, the invariance, the variance, and the covariance terms. The invariance term drives representations to be invariant to input transform, the variance term ensures each dimension of the representation has enough variability, and the covariance term inhibits co-adaptation of dimensions. The proposed objective function shows competitive performance to existing self-supervised learning techniques.

**Summary Of The Review:**

I vote to reject because the contributions of the paper are not well demonstrated in the paper.

---

> ### Author Response · Authors · 2021-11-19
> **Clarification | Difference with Barlow Twins**
>
> 1) *It is unclear that the collapse of representations, the main problem tackled by the paper, is the major bottleneck in self-supervised learning. The experimental results presented in Table 1 and Table 2 are okay, but not pushing the boundary of self-supervised learning.*
>
> We believe that the collapse problem is central self-supervised learning, as all other none-contrastive methods prevent it in a way that is not understood. We explain in KA3G 1) answer and in Appendix B our intuition on why Barlow Twins representations do not collapse. The variance term of VICReg is the only component that ensure a definitive solution against collapse, and we therefore believe that VICReg offers is better formulation. Moreover as we explain in KA3G 2) answer, decoupling the cross-correlation regularization term of Barlow Twins into a covariance and a variance term allows to regularize the two branches of the architecture separately, which is more suited for multi-modal setups and offers better performance in practice in these setups.
>
> 2) *While Table 3 and Table 5 showed that VICReg is more suitable for using heterogeneous encoders, the necessity of heterogeneous encoders is not demonstrated very clearly, because the setting is not practical. The performances reported in Table 3 are far from the state-of-the-art, and in Table 5, the shared weight setting performs best. A more natural setting, such as representation learning for multi-modal data as in VSE [1], should be investigated.*
>
> We agree that the performance of table 3 is not state-of-the-art. The experiment purpose was only to show the better performance of VICReg compared to Barlow Twins in a simple setting. We provided in 8cmN 2) answer a comparison of VICReg and Barlow Twins in the multi-modal setting similar of [1]. VICReg outperforms Barlow Twins and the original contrastive loss proposed in [1]. It greatly benefits from the ability to regularize the two branches separately.
>
> 3) *The contributions of the variance term and the covariance term are not well analyzed. Table 4 is supposed to show the contributions, but it lacks CovReg column so that the conclusion from the table is somewhat vague. Additional efforts for illustrating the effect of the variance and the covariance terms will make the paper more persuasive.*
>
> We provide an experiment where we add covariance regularization to BYOL, SimSiam and Barlow Twins. These methods already decorrelate well their representations, and optimizing the covariance term along with their objective is hard and yield poor performance, which is shown in the table.
>
> |								|	Cov Reg       |
> | ---                          | ---        |
> | BYOL							|   44.3     |
> | SimSiam						|	 31.9     |
> | Barlow Twins 					|   56.8      |
>
> Adding variance regularization back in these settings completely stabilizes the training, as shown in the column Var/Col Reg of Table 4.
>
>
> 4) *The difference from Barlow Twins needs to be elaborated in detail. Otherwise, the proposed method is conceived as a minor improvement over Barlow Twins. I found that the definition of the covariance term is meaningfully different from that of Barlow Twins, but it is not emphasized.*
>
> We make a detailed comparison of the VICReg loss with the Barlow Twins loss in Appendix B. Regarding the covariance term, the main difference is that VICReg regularizes the covariance terms of each branch individually, while Barlow Twins regularizes the cross-correlation of the two branches together. Regularizing the covariance of each branch individually leads to a collapse of the representations, which is prevented by the variance term. Regularizing the cross-correlation does not lead to a collapse, which is a phenomenon that we do not understand. The formulation of VICReg has a clear interpretation, and allows the regularization of each branch separately, which as we explain in KA3G 2) answer, yields better performance in multi-modal settings.

---

### Official Review · Reviewer_8cmN · 2021-10-28

**Correctness:** 4
**Technical Novelty And Significance:** 3
**Empirical Novelty And Significance:** 3
**Recommendation:** 6
**Confidence:** 4

**Main Review:**


Advantages:
1. Authors give an explicit loss function to deal with the collapsed solution problem, which is understandable and explainable compared with BYOL and SimSiam. And the design of minimizing standard deviation for each dimension is insightful.
2. The application of minimizing variance and covariance to other methods, especially SimSiam, is interesting, which can help people understand the mechanism of how negative-free methods work.
3. Well-written and easy to follow.

Comments:
1. The invariance term and covariance term seems a decouple version of BarlowTwins. So I thought the main difference is the variance term.  However from the results, it seems that VICReg does not bring extra improvements compared with BarlowTwins. It is not clear that what kind of problem authors aim to solve. If the variance term is the key, it will be better to show the std of BarlowTwins features, and give more analysis of why the combination of variance-invariance-covariance is advantageous.
2. Authors emphasize that one of the advantages of VICReg is it does not require the weight sharing. It is indeed the VICReg can work without siamese network design, but the property maybe not a exclusive advantage of VICReg. According to my understanding, SimCLR, Barlow Twins can also work with two different architectures.  I thought authors should also compare with these method in the setting of non-shared architectures.
3. About the ESC-50 experiments. It is not clearly that why VICReg perform much better than BarlowTwins in this experiment. And I can not find details in the paper that whether BarlowTwins also use the multi-modal data. Because I believe that Barlow Twins can also work with different architectures, so it is important to figure out why VICReg perform better.
4. Table 4 shows the effect of variance term and covariance term on different method, but missing BarlowTwins. I believe the effect of variance term on BarlowTwins is a key experiment to compare.

**Summary Of The Paper:**

The paper propose a new self-supervised method. New loss is designed to explicitly avoid collapsed solution.

**Summary Of The Review:**

The variance-invariance-covariance framework is insightful, but the experiments are not so convincing.

---

> ### Author Response · Authors · 2021-11-19
> **Clarification | Multi-modal experiment**
>
> 1) *The invariance term and covariance term seems a decouple version of BarlowTwins. So I thought the main difference is the variance term. However from the results, it seems that VICReg does not bring extra improvements compared with BarlowTwins. It is not clear that what kind of problem authors aim to solve. If the variance term is the key, it will be better to show the std of BarlowTwins features, and give more analysis of why the combination of variance-invariance-covariance is advantageous.*
>
> As explained in KA3G 1) answer, VICReg should not be seen as Barlow Twins with an additional variance term, but rather as a new formulation that prevents both a representational and an informational type of collapse, by construction of the loss function. VICReg without the variance term completely collapses, and as explained in Appendix B, the reason why Barlow Twins does not collapse is still a mystery which might be related to the normalization done before the loss. VICReg has a clear interpretation and prevents collapse by design of its loss function.
> Decoupling covariance and variance in two separate terms allows to regularise the branches separately, which as we show experimentally is key perform better in multi-modal setups.
>
> 2) *Authors emphasize that one of the advantages of VICReg is it does not require the weight sharing. It is indeed the VICReg can work without siamese network design, but the property maybe not a exclusive advantage of VICReg. According to my understanding, SimCLR, Barlow Twins can also work with two different architectures. I thought authors should also compare with these method in the setting of non-shared architectures.*
>
> Table 5 provide an experimental comparison of VICReg against Barlow Twins and SimCLR in scenarios where the branches dot not share their architecture. In the last column, a ResNet-50 is used for one branch and a vision transformer (ViT-S) is used in the second branch. As explained in KA3G 2) answer, VICReg is more robust than Barlow Twins in these setups, and offers better performances than SimCLR.
>
> We provide an additional experiment where we jointly pretrain on images and corresponding captions on the MS-COCO dataset, and evaluate on image-to-text and text-to-image retrieval tasks on the 5k test evaluation protocol.  We compare VICReg to Barlow Twins, and to a contrastive loss used in VSE++ (See [1] of dMLo review), using the same setting and encoder architecture as the original setting in [1]. We tune the coefficient of the variance and covariance terms separately for each branches, which is only possible with VICReg.
>
>
> |	MS-COCO 5k-test				  | Image-to-text             |||        Text-to-Image           |||
> | ---                            |    ---  |    ---  | ---     |  ---    |  ---   | ---         |
> |				                 |	R1	  |	R5	     | R10   |   R1    |  R5    | R10             |
> | Contrastive (VSE++)	         |	30.3  |  59.4   |  72.4  |   41.3  | 71.1   |   81.2         |
> | Barlow Twins		             |	31.4  | 60,4    |  75.1  |   42.9  |  74.0  |  83.5          |
> | VICReg			             |	33.6  | 62.7    |  77.9  |   45.2  |  76.1  |  84.2          |
>
> VICReg outperforms both the contrastive loss and the Barlow Twins loss by a significant margin.
>
>
>
> 3) *About the ESC-50 experiments. It is not clearly that why VICReg perform much better than BarlowTwins in this experiment. And I can not find details in the paper that whether BarlowTwins also use the multi-modal data. Because I believe that Barlow Twins can also work with different architectures, so it is important to figure out why VICReg perform better.*
>
> We explain in KA3G 2) answer, our intuition on why VICReg performs better than Barlow Twins in these none-shared weights scenarios. The fact that VICReg regularises its two branches separately, by using the covariance matrices of each branch and not the cross-correlation matrix is key, as it allows the branches to be regularized by taking into account the statistics of its outputs. We validate this intuition with our additional multi-modal experiment on image and text pretraining, where VICReg performs better than Barlow Twins, and than a contrastive loss.
>
> 4) *Table 4 shows the effect of variance term and covariance term on different method, but missing BarlowTwins. I believe the effect of variance term on BarlowTwins is a key experiment to compare.*
>
> Adding variance regularization in Barlow Twins does not yield any performance improvement, as it is already able to prevent collapse and maintain the variance of its representations in a mysterious way. We explain our intuition on why this is the case in KA3G 1) answer as well as in Appendix B. Our variance term is the only component among none-contrastive self-supervised methods that prevents collapse by design and has a clear interpretation.

---

### Official Review · Reviewer_QN96 · 2021-10-31

**Correctness:** 3
**Technical Novelty And Significance:** 3
**Empirical Novelty And Significance:** 3
**Recommendation:** 6
**Confidence:** 4

**Main Review:**

Strengths:

1. The paper is well-written and easy to follow;

2. The method is simple and achieve comparable performance for both linear evaluation and downstream transferring;

3. The authors provide a clear and detailed discussion to compare this work with the previous methods.


Weaknesses:

1. The reviewer does not feel very excited about the work. In fact, the three loss functions are not very novel. As the reviewer mentioned in the summary, the covariance term is just directly modified from the Barlow Twins. The same measure of the variance term has been used in some previous works (e.g., SimSiam) to analyze the model collapse problem, while it is not designed as a pre-trained loss function.

2. In the table 1, the comparison with previous methods might not be very fair. In particular, some compared methods such as MoCo v1/v2, SimSiam and InfoMin are just pre-trained for 800 epochs, while the proposed model is pre-trained for 1000 epochs. Besides, some of the previous methods do not use LARS optimizer and warmup strategy that are applied in this work.

3. While the proposed method is simple, however, the computation time of the covariance matrix is quadratic in terms of the feature dimension, which slow the pre-training significantly.

4. Although the authors have provided detailed discussions to illustrate the differences of this work with previous works in terms of the design details, however, can the authors elaborate theoretically on the advantages of the variance and covariance terms against the whitening operation in W-MSE?

5. Besides ResNet-50, it will be more beneficial to the community if the authors can compare the proposed method with the MoCo v3, by showing the performance with the Transformer backbone.

**Summary Of The Paper:**

This paper combines three objective functions for the self-supervised visual pre-training on ImageNet.

(1) The alignment between the two different views of an identical image, which is very common for existing methods;

(2) The covariance term to bring the off-diagonal coefficients of the features' covariance matrix to zero, which is modified from the Barlow Twins;

(3) The variance term that defines a hinge function on the standard deviation of embeddings along the batch dimension for every specific dimension of the feature projections . To the best of the reviewer's knowledge, such objective function is firstly applied for the visual pre-training in this paper, although the same measure has been used to analyze the model collapse problem (e.g., in the paper of SimSiam), but not be designed as a specific pre-trained loss function.

**Summary Of The Review:**

Overall, the reviewer tend to vote for accept for this work since the proposed method is simple and it has conducted thoughtful experiments to demonstrate the effectiveness.

The reviewer encourages the authors to speed up the proposed method, make the comparison with previous methods fairer and try to test the method on different architecture.

---

> ### Author Response · Authors · 2021-11-19
> **Fairer comparison | VICReg with transformers**
>
> 1) *As the reviewer mentioned in the summary, the covariance term is just directly modified from the Barlow Twins. The same measure of the variance term has been used in some previous works (e.g., SimSiam) to analyze the model collapse problem, while it is not designed as a pre-trained loss function.*
>
> As explained in KA3G 1) answer, VICReg propose to put these losses together in a framework that is designed to prevent collapse by construction, and that offers additional benefits in multi-modal setups. We believe that using the variance as a pre-trained loss function is new and is a contribution in itself.
>
> 2) *In the table 1, the comparison with previous methods might not be very fair. In particular, some compared methods such as MoCo v1/v2, SimSiam and InfoMin are just pre-trained for 800 epochs, while the proposed model is pre-trained for 1000 epochs. Besides, some of the previous methods do not use LARS optimizer and warmup strategy that are applied in this work.*
>
> In order to make a fair comparison, we train VICReg on 800 epochs with a standard SGD optimizer without warmup on 800 epochs. We also train SimSiam and MoCo on 1000 epochs using the LARS optimizer and a warmup strategy. We present the results in the following table:
>
> | 				| 800 ep / SGD / no warmup | 1000 ep / LARS / warmup |
> | ---          | ---                         | ---                     |
> | VICReg		|	71.9					|	 73.2   |
> | SimSiam	|		71.3				|		 70.6  |
> | MoCO		|	71.1					|	 71.8     |
>
> We see that VICReg benefits the most from the LARS optimizer and the warmup strategy.
>
> 3) *While the proposed method is simple, however, the computation time of the covariance matrix is quadratic in terms of the feature dimension, which slow the pre-training significantly.*
>
> We report in Appendix E the running time and peak memory of VICReg compared to other self-supervised methods. VICReg offers a good tradeoff in terms of performance and computational ressources required when compared to other methods such as SwAV and BYOL. The time required for the covariance matrix computation is negligible compared to the time required by the encoder.
>
> 4) *Although the authors have provided detailed discussions to illustrate the differences of this work with previous works in terms of the design details, however, can the authors elaborate theoretically on the advantages of the variance and covariance terms against the whitening operation in W-MSE?*
>
> W-MSE aims to decorrelates the embeddings in order to maximise the informational content and prevent collapse, which is the same objective as VICReg. As explained in Appendix E, computing the whitening operator of W-MSE requires to inverse the covariance matrix which is computationally costly and very unstable. In comparison VICReg uses the covariance matrix and achieve the same purpose without requiring to inverse it.
>
> 5) *Besides ResNet-50, it will be more beneficial to the community if the authors can compare the proposed method with the MoCo v3, by showing the performance with the Transformer backbone.*
>
> We provide an additional experiment where we use a vision transformer backbone, and compare with MoCo v3. VICReg performs slightly worse than MoCo v3 but still achieves competitive results using a vision transformer architecture.
>
> |					|		R50	(800ep)		| ViT-S (300ep) |  ViT-B (300ep) |
> |           ---    |            ---       | ---           | ---            |
> | VICReg 			|			73.2		|	72.1        |      76.2      |
> | MoCo v3			|			73.8		|	72.5        |      76.7      |

---

### Official Review · Reviewer_KA3G · 2021-11-02

**Correctness:** 3
**Technical Novelty And Significance:** 2
**Empirical Novelty And Significance:** 3
**Recommendation:** 6
**Confidence:** 3

**Main Review:**

Strengths:
+ The authors did a very good job in explaining the background and presenting the paper. The main idea is conveyed very clearly.
+ The idea of adding a variance term to the total loss to avoid representation collapse is interesting, intuitive and novel.
+ A great number of experiments compared with prior methods with detailed set up have been conducted.
+ Ablation analysis has also been conducted, showcasing the effects of different components.
+ A study on multi-modal signal representation learning is presented, demonstrating the importance of not requiring architecture or weight sharing in two branches.

Weakness:
- It seems that the main contribution, which is the variance term, plays a somewhat insignificant role in Table 1 and Table 2. In fact, compared to Barlow Twins, which does not have the variance term, the proposed method in many cases actually underperforms.
- Not requiring shared weight between different branches is a feature of Barlow Twin as well. Can the authors provide an explanation on the inferior performance of Barlow Twin in Table 3 and Table 5?
- The authors mentioned that using standard deviation instead of the variance in the hinge loss is important. Can a toy numerical example be provided to showcase the presence of representation collapse when variance is used?

**Summary Of The Paper:**

The authors propose a Variance-Invariance-Covariance regularization technique for self-supervised learning. The loss function used in the paper consists of three terms: the invariance term encouraging samples with different view to have similar embedding; the variance term, which is a hinge loss on the variance of the embedded variables (this is the main contribution of the paper, and the authors claim that it helps to avoid variance collapse); and a covariance term which borrows from the previous work Barlow Twin. The proposed method has greater flexibility for siamese architecture design, such as not requiring batch-normalization and weight-sharing, which the authors claim opens the door for multi-modal signal embedding. Experiments and ablation study have been conducted to demonstrate the performance of the proposed components.

**Summary Of The Review:**

The paper is easy to understand, and has its contribution and novelty. Many experiments have been conducted, but theory is a bit lacking. I am willing to increase my rating if the authors can respond to my comments.

---

> ### Author Response · Authors · 2021-11-19
> **VICReg's clear formulation | Better multi-modal performance**
>
> 1) *It seems that the main contribution, which is the variance term, plays a somewhat insignificant role in Table 1 and Table 2. In fact, compared to Barlow Twins, which does not have the variance term, the proposed method in many cases actually underperforms.*
>
> VICReg should not be seen as Barlow Twins with an additional variance term, but rather as a new formulation that prevents both a representational and an informational type of collapse, by construction of the loss function. As explained in Appendix B, there is theoretically nothing preventing the embeddings of Barlow Twins to shrink and become constant to numerical precision, and this phenomenon is avoided by adding a constant scalar in the denominator of the normalization term before the Barlow Twins loss. Our experiments show that this normalization is crucial and that Barlow Twins partially collapses without it (The final performance on linear classification drops from 71.4 to 53.4, and the variance of the final representation drops significantly). We believe that VICReg is a better formulation and a definitive solution against the collapse problem.
>
> 2) *Not requiring shared weight between different branches is a feature of Barlow Twin as well. Can the authors provide an explanation on the inferior performance of Barlow Twin in Table 3 and Table 5?*
>
> One fundamental difference of VICReg compared to Barlow Twins is the way the branches are regularized. In VICReg, both branches are regularized independently, as the covariance term is applied on each branch separately, which works better in the scenarios where the branches are completely different, have different types of architecture and process different types of data. Indeed the statistics of the representation of these two branches can be very different, and the amount of regularization required for each may vary a lot. In Barlow Twins, the regularization is applied on the cross-correlation matrix, which favors the scenarios where the branches produce outputs with similar statistics. We provide in 8cmN 2) answer a new multi-modal experiment where we pretrain on pairs of text and images data. We regularize each branch with a different coefficient, which is not possible with Barlow Twins, and we show that VICReg outperforms Barlow Twins on image and text retrieval downstream tasks. We believe that this separation is the key ingredient that make VICReg work better in Table 3 and 5 as well.
>
> 3) *The authors mentioned that using standard deviation instead of the variance in the hinge loss is important. Can a toy numerical example be provided to showcase the presence of representation collapse when variance is used?*
>
> In our experiments with the variance instead of the standard deviation, the standard deviation of the representations projected on the unit sphere during training, which is a metric we use to measure diversity within the representations, as well as the loss, slowly fall to 0 (in few iterations). This shows that the representations collapse to a single 0 vector. We give a simple theoretical explanation in section 4 of the paper on why the representations collapse.

---

> > ### Comment · Reviewer_KA3G · 2021-11-25
> > **Thank you for the response.**
> >
> > I want to thank the author for the detailed response. Even though the performance gain (or lack thereof) of VICReg in Table 1 and 2 is somewhat insignificant, I think the authors have answered most of my concerns. I am willing to raise my score to marginally above the acceptance threshold.

---

### Decision · Program_Chairs · 2022-01-20

**Decision:**

Accept (Poster)

**Comment:**

This paper presents a self-supervised learning method for the multi-modal setting where each modality has its own feature extraction mapping, and i) the extracted features shall be close for paired data,  ii) in the feature space each view has close to diagonal covariance, while iii) the scale for each feature dimension is constrained away from zero to avoid trivial features. The presentation is clear and the reviewers do not have major confusion on the methodology. There have been some discussions between the authors and reviewers, and most questions on the empirical study have been addressed by the authors with additional experiments. The remaining concern is on the novelty (difference from prior SSL methods especially Barlow-Twins) and significance.  I think that while it is relatively straightforward to extend methods like Barlow-twins to the multi-modal setting, I do see the value of empirically demonstrating the effectiveness of an alternative loss to the currently pervasive contrastive learning paradigm, and hence the paper is worth discussion in my opinion. In the end, the method resembles classical multi-modal methods like canonical correlation analysis, in terms of the objective (matching paired data in latent space) and constraints (un-correlated feature in each view, and unit-scale constraint for each feature dimension); such connections shall be discussed.